# Unveiling the Chemistry and Synthetic Potential of Catalytic Cycloaddition Reaction of Allenes: A Review

**DOI:** 10.3390/molecules28020704

**Published:** 2023-01-10

**Authors:** Sana Sikandar, Ameer Fawad Zahoor, Abdul Ghaffar, Muhammad Naveed Anjum, Razia Noreen, Ali Irfan, Bushra Munir, Katarzyna Kotwica-Mojzych, Mariusz Mojzych

**Affiliations:** 1Department of Chemistry, Government College University Faisalabad, Faisalabad 38000, Pakistan; 2Department of Biochemistry, Government College University Faisalabad, Faisalabad 38000, Pakistan; 3Department of Applied Chemistry, Government College University Faisalabad, Faisalabad 38000, Pakistan; 4Institute of Chemistry, University of Sargodha, Sargodha 40100, Pakistan; 5Laboratory of Experimental Cytology, Medical University of Lublin, Radziwiłłowska 11, 20-080 Lublin, Poland; 6Department of Chemistry, Siedlce University of Natural Sciences and Humanities, 3-go Maja 54, 08-110 Siedlce, Poland

**Keywords:** cycloaddition reactions, metal catalysis, allenes, bioactive molecules

## Abstract

Allenes with two carbon–carbon double bonds belong to a unique class of unsaturated hydrocarbons. The central carbon atom of allene is sp hybridized and forms two σ-bonds and two π-bonds with two terminal sp^2^ hybridized carbon atoms. The chemistry of allenes has been well documented over the last decades. They are more reactive than alkenes due to higher strain and exhibit significant axial chirality, thus playing a vital role in asymmetric synthesis. Over a variety of organic transformations, allenes specifically undergo classical metal catalyzed cycloaddition reactions to obtain chemo-, regio- and stereoselective cycloadducts. This review briefly describes different types of annulations including [2+2], [2+2+1], [3+2], [2+2+2], [4+2], [5+2], [6+2] cycloadditions using titanium, cobalt, rhodium, nickel, palladium, platinum, gold and phosphine catalyzed reactions along with a mechanistic study of some highlighted protocols. The synthetic applications of these reactions towards the synthesis of natural products such as aristeromycin, *ent*-[3]-ladderanol, waihoensene(−)-vindoline and (+)-4-*epi*-vindoline have also been described.

## 1. Introduction

The carbon–carbon bond formation by various chemical processes is extremely important in organic chemistry, especially when cyclic systems with complex structures are generated from simple precursors [1,2,3,4,5]. Cycloaddition reactions play a pivotal role in this regard for the synthesis of a number of heterocyclic molecule systems with high yield. Moreover, they proceed with chemo-, regio- and stereoselectivity and thus attracting a great attention of organic chemists. A major part of a literature review in organic chemistry highlights the latest discoveries, shedding new insights on synthetic and mechanistic aspects of cycloaddition processes [6,7,8]. Cycloaddition reactions are generally single-step reactions which occurs on joining two π-systems at their ends forming a cyclic compound through formation of two sigma bonds, while each reactant loses one π-bond in the process [9]. However, recently, there have been various attempts made on the postulation of the step-wise mechanism of cycloaddition reactions, specifically Diels–Alder reactions [10]. They are proposed to proceed via zwitterionic or biradical intermediates [11]. Moreover, these reactions are not only important for simple organic molecule synthesis but are also vital for the modern synthesis of natural products as well as biologically active substances [12,13,14,15]. Metal catalysts in these reactions also enhance the selective formation of several stereocenters and their integration in target molecules [16].

All allenes whether synthetic intermediates or in natural products are based on a 1,2-propadiene structure. Their synthetic origin traces back to 1887. Allenes though devoid of chirality are useful for synthesizing chiral compounds. Their significant applications in organometallic chemistry is also well documented [17,18,19,20]. Allenes and their derivatives reacting with various unsaturated compounds via cycloaddition reactions are involved in the synthesis of indole, pyridine, furan and other cyclic compounds. In addition to this, their symmetry, isomeric properties and characteristic reactivity (with nucleophiles, electrophiles and radical species) have fascinated researchers in the recent past to explore wide open possibilities to discover various building blocks required for the construction of biologically active materials through a variety of cycloaddition reactions of allenes including [2+2], [2+2+1], [3+2], [2+2+2], [4+2], [3+2+2], [5+2] and [6+2] [21,22,23,24,25,26]. All data related to transition metal catalyzed and phosphine catalyzed cycloaddition reactions of allenes, investigated since 2015, are represented in this review. Moreover, synthetic applications of these reactions towards the synthesis of natural products are also highlighted.

## 2. Review of Literature

### 2.1. Transition-Metal Catalyzed Cycloaddition Reactions of Allenes

#### 2.1.1. Titanium Catalyzed Reactions

A report on the preparation of bicyclonona-2,4-dienes and bicyclonona-2,4,7-trienes via [6+2] cycloaddition of allenes and alkynes with 7-substituted 1,3,5-cycloheptatrienes catalyzed by titanium was reported by D’yakonov and co-workers [27]. Ti(acac)_2_Cl_2_-Et_2_AlCl was utilized as an effective catalyst to obtain the desired products in highest yield. For example, bicyclonona-2,4-diene **3** was synthesized in 90% yield via [6+2] cycloaddition of allene **2** with 7-*n*butyl-1,3,5-cycloheptatriene **1** at 80 °C (Figure 1).

#### 2.1.2. Cobalt-Catalyzed Reactions

Ding and Yoshikai reported cobalt-catalyzed intermolecular [2+2] cycloaddition of allenes and alkynes to synthesize several 3-alkylidenecyclobutenes [28]. 4-Alkylidenecobaltcyclopentene formed as intermediate via oxidative cyclization preceded by C-C reductive elimination to form desired 3-alkylidenecyclobutenes in good yields (up to 94%) and high regioselectivities. For example, compound **6** was obtained in highest yield (94%) from cycloaddition of alkyne **4** and allene **5** in the presence of CoBr_2_ (5 mol%), dppf (5 mol%) and In (20 mol%) (Figure 2).

Yoshikai and co-workers developed a simple and efficient method for the preparation of 3-alkylidenecyclopentanol derivatives by cobalt-catalyzed cycloaddition reaction [29]. Several monocyclic and fused polycyclic 3-alkylidenecyclopentanols with high regio- and diastereoselectivities were formed in low to high yields (21–91%) via [3+2] cycloaddition of cyclopropanols with allenes using cobalt(ΙΙ) catalyst, diphosphine ligand and amine base. Bicyclic cyclopentanol **9** was obtained in 90% yield from 1-(4-chlorophenyl) cyclopropanol (R = Cl) **7** and cyclonona-1,2-diene **8**, while the highest yield (91%) of 3-alkylidenecyclopentanol **10** as single diastereomer was obtained using 1-phenylcyclopropanol (R = H) **7** and monosubstituted allene **11** in a [3+2] cycloaddition reaction using CoI_2_ (10 mol%), dppm (10 mol%) and DABCO at 80 °C (Figure 3).

#### 2.1.3. Rhodium Catalyzed Reactions

Casanova et al. synthesized 2H-chromene derivatives via Rh-catalyzed [5+1] annulation of several allenes with 2-alkenylphenols [30]. The synthetic route was accomplished by rhodium catalysis in which allenes participating as a one-carbon cycloaddition partner and breaking of a C-H bond of 2-alkenylphenols resulted in 2,2-disubstituted 2H-chromenes. Products **14** and **16** were obtained in highest yield (98%) using [{Cp*RhCl_2_}_2_] (2.5 mol%) and copper(ΙΙ) acetate monohydrate at 85 °C via oxidative [5+1] annulation of 2-alkenylphenol **12** with vinylidinecyclohexane **13** and allenyl alcohol **15**, respectively. The mechanism of this protocol starts from the intramolecular coordination of phenol with rhodium(III) complex, which gives rhodacycle after rearomatization (**B**). This intermediate first coordinates with allene and then undergoes *ꞵ*-hydride elimination. The next step involved “1,7-H-shift”, and 6π-electrocyclic reaction to afford a targeted chromene derivative (Figure 4).

Cancer is one of the major medical challenges to mankind. 2,6-Naphthyridine gained significant attention in medicinal chemistry due to the diverse bioactivities and is currently under consideration for cancer and HIV research [31]. Efficient access to 2,6-naphthyridine derivatives was achieved via Rh-catalyzed [2+2+2] addition of cyano-yne-allene by Haraburda et al. [32]. Their synthetic methodology involved the intramolecular cycloaddition of external bonds of allene in the presence of the rhodium catalyst that first resulted in unsaturated pyridines and gave 2,6-naphthyridine after dehydrogenation. For example, when N-tosyl (NTs) containing cyano-yne-allene **17** was treated with Wilkinson’s catalyst (10 mol%) in the presence of 0.05 mol% Et_3_N under microwave irradiation, tricyclic adduct **18** was formed with a 66% yield, which after aromatization, resulted in 2,6-naphthyridine **19** (74%) (Figure 5). Later on, the same research group reported a [2+2+2] cycloaddition reaction of allene-yne-allene and allene-ene-allene linked with N-tosyl to obtain corresponding fused polycycles in a stereoselective manner [33]. Wilkinson’s catalyst was used for this purpose and as a result, high selectivity was obtained.

An effectual synthesis of enantioenriched pyrrolidine derivatives via Rh-catalyzed regiodivergent intramolecular [3+2] cycloaddition of allenes with vinyl aziridines was revealed by Lin et al. [34]. 2-Methylene-pyrrolidines were obtained by the [3+2] cycloaddition of distal C-C double bond of allene with vinyl aziridine, while the [3+2] cycloaddition of proximal carbon–carbon double bond of N-allenamides with vinyl aziridines resulted in 3-methylene-pyrrolidines. Noticeably, 3-methylene-pyrrolidines were formed in the presence of [Rh(NBD)_2_]^+^BF_4_^−^ in DCE at 0 °C, while 2-methylene-pyrrolidines were obtained using [{Rh(NBD)Cl}_2_]/AgOTf as catalyst in acetone at 0 °C. Among others, 2- and 3-methylene-pyrrolidines **22** and **24** were synthesized in high yields (95% and 94%, respectively) and excellent enantioselectivities (97% and 93% *ee*, respectively) via cycloaddition of allenamide **21** and allene **23** with vinyl aziridine **20** by loading 5 mol% of [Rh(NBD)_2_]^+^BF^−^_4_ and [{Rh(NBD)Cl}_2_]AgOTf, respectively (Figure 6).

A rhodium(Ι)-catalyzed formation of several bicyclo [3.3.0]octanes was reported by Liu and Yu via intramolecular [3+2] cycloaddition of trans-2-allene-vinylcyclopropanes in moderate to high yields (62–80%) [35]. Among different NTs-, NNs-, NSO_2_Ph- and NBs-tethered substrates, NNs bearing trans-2-allene-vinylcyclopropane **25** gave the highest yield (80%) of bicyclo [3.3.0]octane **26** using 5 mol% Rh(CO)PMe_3_)_2_Cl and 5 mol% of AgOTf (Figure 7).

A Rh-catalyzed intramolecular [4+2] cycloaddition of allene-1,3-diene to afford *cis*-6,5-fused bicycles with high diastereoselectivities was reported by Han and Ma [36]. Synthesis of *cis*-6,5-fused bicycles involved (1) cyclometalation, (2) allylic rearrangement, and (3) reductive elimination. They found that the configuration of the non-bridging tertiary carbon was directed by the configuration of C-C double bond in 1,3-diene. For example, diastereoisomer (3aR*,5R*,7aR*,*Z*)-**28** was synthesized in 83% yield via [4+2] cycloaddition of **27** (C=C (2*E*,4*Z*)) using 2 mol% RhCl(PPh_3_)_3_ at 80 °C (Figure 8).

Another report on the chemo- and diastereoselective synthesis of *cis*-fused [3.4.0]-bicycles having three chiral centers via intramolecular [4+2] cycloaddition of optically active chiral allenes-1,3-dienes by rhodium catalysis was reported by Ma and co-workers [37]. Among different *cis*-fused [3.4.0]-bicycles, (3a*R*,5*R*,7a*R*,*E*)-**30** was obtained in highest yield (82%) with good enantioselectivity (94% *ee*) from intramolecular [4+2] cycloaddition of malonate tether (R_a_,2*E*,4*E*)-**29** (94% *ee*) using RhCl(PPh_3_)_3_ (3 mol%) and AgSbF_6_ (5 mol%) (Figure 9). 

Schomaker and co-workers described the formation of functionalized aminated cycloheptenes as well as cycloheptanes via sequential tendem allene aziridination/intermolecular [4+3] cycloaddition/reduction [38]. An intermolecular, stereodivergent [4+3] cycloaddition occurred through 2-amidoallyl cations formation from substituted allenes that resulted in the formation of all four diastereoisomers by *endo* cyclization. Among several products, **32** was synthesized in highest yield (62%) with good diastereoselectivity (19:4:1:1) via Rh(Ι)-catalyzed allenic aziridination of **31** and [4+3] cycloaddition of furan in MeNO_2_ followed by reduction and hydrogenation (Figure 10). 

An effective protocol to synthesize 5–7 fused bicyclic compounds was developed by Tang and co-workers [39]. 3-Acyloxy-1,4-enynes (ACEs) successfully underwent intramolecular [5+2] cycloaddition with alkene or allene to synthesize bicyclic products using Rh-catalyst and phosphine ligand with high diastereoselectivity. Both *cis*-**34** and *trans*-**34** isomer were formed from intramolecular [5+2] cycloaddition of **33** bearing *gem*-dimethyl groups in the linker region that gave 58% highest yield while only *cis*-**35** formed in the absence of *gem*-dimethyl groups from **33** bearing electron-rich dimethylaminobenzoate ester. The mechanism is presented in Figure 11.

Liu and Yu reported the preparation of bicyclo [4.3.1]decane skeleton using the same [5+2] cycloaddition strategy by replacing ene with allene in *cis*-ene-VCPs, the inner double bond of which acted as 2π component [40]. The highest yield of product **37** and **38** (80%) was obtained using *cis*-allene-VCP **36** having methyl and ethyl groups in the presence of [Rh(CO)_2_Cl]_2_ (5 mol%) catalyst at 80 °C (Figure 12).

Guaianolides (sesquiterpene lactones) are biologically important scaffolds as they possess many activities such as antitumor and anti-inflammatory activities [41]. Wells and Brummond reported the preparation of bicyclo [5.3.0]decadienones via rhodium(Ι)-catalyzed [2+2+1] cycloaddition of methyl substituted allenes with alkynes [42]. They first prepared allene-ynes starting from allenes in the following steps: (1) reduction with LiAlH_4_ and then mesylation of resulting hydroxyl group, (2) treatment with sodium triethyl methanetricarboxylate followed by decarboxylation, (3) deprotonation of malonate derivative, and (4) addition of 1-bromo-2-butyne. These allene-ynes were then transformed to bicyclodecadienones in excellent yields; for example, compound **40** was synthesized in 80% yield from allene-yne **39** by employing rhodium-catalyzed allenic Pauson-Khand reaction (APKR) conditions (Figure 13).

A Rh(Ι)-catalyzed intramolecular [2+2+2] cycloaddition of allenes, alkynes and tethered imines was reported by Oonishi et al. to synthesize fused cyclic amides and 8-azabicyclo octane derivatives [43]. A highly strained intermediate azarhodacycle **42** was formed which gave 5,7-fused cyclic amides and 8-azabicyclo [3.2.1]octanes via reductive elimination in good yields. For example, fused bicyclic amide **43** and 8-azabicyclo [3.2.1]octane **44** were obtained from substrate **41** in 65% and 84% yields using [Rh(dppp)]ClO_4_ (5 mol%) and [Rh(BINAP)]ClO_4_, respectively (Figure 14).

Tanaka and co-workers described the cross-cyclotrimerization and dimerization of alkynes with allenes by rhodium catalysis to afford 3,6-dialkylidenecyclohex-1-enes and substituted dendralenes, respectively in good yields [44]. Two molecules of allenes underwent cross-cyclotrimerization with one molecule of alkyne using [Rh(cod)_2_]BF_4_ as pre-catalyst and *bis* (diphenylphosphino)-binaphthyl (BINAP) as ligand to synthesize several 3,6-dialkylidenecyclohex-1-ene derivatives. Cross-dimerization products were obtained by the reaction of alkynes with di- or tri-substituted allenes via *β*-H elimination from rhodacycles. The highest yield (70%) of 3,6-dialkylidenecyclohex-1-enes **47** and **48** (88:12) was obtained with alkyne **46** while substituted alkyne **49** reacted with tri-substituted allene **45** to afford substituted dendralene **50** in 89% yield (Figure 15). Formation of **46** and **49** proceeded through the synthesis of rhodacyclopentene as a result of the reaction of allene, alkyne and rhodium catalyst. Insertion of second allene **45** and then reductive elimination resulted in the formation of compound **47**, while *β*-H elimination from rhodocyclopentene and subsequent reductive elimination furnished compound **50**.

Rhodium-catalyzed synthesis of [4.2.1]-bicyclic compounds with two quaternary carbons was reported by Zhou and Dong [45]. An intramolecular [4+1] cyclization of allenes with cyclobutanones was achieved in which allene acted as a one-carbon unit and a reaction proceeded by carbon–carbon activation of cyclobutanones. A wide range of fused/bridged bicycles was formed, but 6–5 bridged bicycle **52** was synthesized in highest yield (96%) from allene **51** using [Rh(C_2_H_4_)_2_Cl]_2_ (5 mol%) catalyst and P(3,5-C_6_H_3_(CF_3_)_2_)_3_ (24 mol%) ligand at elevated temperature 150 °C (Figure 16).

Zhao et al. reported the preparation of alkylidene tetralins with two adjacent stereogenic carbons from Rh-catalyzed [4+2] cycloaddition of allenes into benzocyclobutenols [46]. Reaction conditions were optimized to achieve best results for the construction of alkylidene tetralins and it was found that the highest yield was obtained using 2 mol% [Rh(cod)(OH)]_2_ as catalyst at 100 °C in toluene. A wide range of benzocyclobutenols and allenes underwent [4+2] cycloaddition to afford various alkylidene tetralins but **55** was obtained in an excellent yield (93%) from benzocyclobutenols **53** and allene **54** with high diastereoselectivity (>19:1) containing *p*-tolyl and *p*-anisyl substituents (Figure 17).

Rh-catalyzed [4+2+1] cycloaddition reaction of in situ generated ene/yne-ene-allenes with CO to synthesize seven-membered carbocyclic compounds fused with five-membered rings was first published by Yu and co-workers [15]. Ene/yne-ene-allenes were generated from ene/yne-ene-propargyl esters via 1,3-acyloxy migration that underwent cyclization (oxidative), alkene/alkyne insertion followed by CO insertion and reductive elimination. The highest yielded bicyclic 5/7 compound **57** (94%) was synthesized from [4+2+1] cycloaddition of propargyl ester **56** in reaction conditions of 1 atm CO using [Rh(COD)Cl]_2_ (5 mol%) in DCE (Figure 18).

Mukai and co-workers prepared 1,5,6,7-tetrahydroazulene skeletons via intramolecular [5+2−2] cycloisomerization of several allene-allenylcyclopropanes by rhodium catalysis [47]. Their synthetic protocol involved the liberation of ethylene from cyclopropane ring that acted as C_1_ building block. Several 1,5,6,7-tetrahydroazulene compounds were synthesized along with cyclopentenylidene derivatives, for example 1,5,6,7-tetrahydroazulene derivative **59** (41%) was obtained from allene-allenylcyclopropane **58** bearing phenylsulfonyl groups on allenyl functionalities, along with the formation of allene **60** (Figure 19).

An efficient synthesis of fused-tricyclic ring systems was reported via the Rh(Ι)-catalyzed [2+2+2] cycloaddition of N-tosyl-tethered allene-(*E*)-ene-ynes by Cassú et al. [48]. The exocyclic double bond in fused-tricycle was chemoselectively formed by the reaction of a proximal double bond of allene. Several allene-ene-yne substrates were prepared from N-tosylallenes and bromoallyls by nucleophilic substitution using K_2_CO_3_ as base and then employed in a [2+2+2] cycloaddition reaction. A highest yield of fused-tricyclic diastereoisomers *syn*-**62** and *anti*-**62** (88%, *syn*:*anti* = 9:1) was obtained using NTs-tethered allene-(*E*)-ene-yne **61** bearing isopropyl group using [RhCl(PPh_3_)_3_] (10 mol%) at 100 °C (Figure 20).

#### 2.1.4. Nickel Catalyzed Reactions

Noucti and Alexanian reported a straightforward and effective approach towards the formation of fused cyclobutanes using an inexpensive first-row catalyst [49]. Several fused cyclobutane derivatives were synthesized via a nickel-catalyzed [2+2] cycloaddition of ene-allenes using phosphine ligand. For example, **64** was synthesized in highest yield (95%) from 1,3-disubstituted allene **63** using [Ni(cod)_2_] (10 mol%) and *bis*(diphenylphosphino)ferrocene (dppf) (10 mol%) at high temperature (100 °C) (Figure 21).

#### 2.1.5. Palladium Catalyzed Reactions

In contrast to the construction of 2H-chromenes via the reaction of allenes with 2-alkenylphenols, Gulìas and co-workers carried out the synthesis of benzoxepines via the Pd(ΙΙ)-catalyzed [5+2] annulation of allenes with *ortho*-alkenylphenols under oxidative conditions [50]. A variety of benzoxepines was obtained by the reaction of readily available 2-alkenylphenols and allenes using catalytic amount of Pd(ΙΙ) and Cu(ΙΙ); however, benzoxepine **67** bearing electron withdrawing substituent at *para* carbon was obtained in highest yield (97%) from *ortho*-alkenylphenol **65** and allene **66**. Computational studies showed that the geometry of metal catalysts (square planar in case of palladium) determined the reaction outcome. A plausible mechanism of this protocol starts from the exchange of ligand between phenol substrate and palladium acetate that generates intermediate (**B**) after the intramolecular reaction of alkene with palladium. This intermediate, after coordination with allene followed by migratory insertion and reductive elimination reaction, gave the desired benzoxepine product (Figure 22).

Mascareñas and co-workers published another report on the Pd-catalyzed formal [5+2] cycloaddition of allenes [51]. They reported the formation of 2,3-dihydro-1H-benzo[*b*]azepines via the [5+2] annulation of allenes with 2-alkenyltriflylanilides using a catalytic amount of Pd(ΙΙ) and Cu(ΙΙ). Among different substituted allenes, 2-vinylidenecyclohexane **13** was found to be highly reactive with 2-alkenylanilide **68** bearing electron acceptor CF_3_ group to give 2,3-dihydrobenzoazepine **69** with 92% yield using 5 mol% Pd(OAc)_2_ and Cu(OAc)_2_·H_2_O. Density functional theory (DFT) calculations showed that the synthesis of benzazepines took place through the C-H activation of 2-alkenyltriflylanilides that involved a metalation–deprotonation (CMD) mechanism (Figure 23).

Vidal et al. described that benzyl and allyltriflimides successfully underwent oxidative [4+2] cycloaddition with allenes using Pd-catalyst to afford tetrahydroisoquinoline and dihydropyridine derivatives [52]. N-benzyltriflimides **72** and N-allyl amines **74** were used in Pd-catalyzed annulation with substituted allenes **70** to synthesize tetrahydroisoquinoline **73** (91% yield) and dihydropyridine **75** (90% yields) in the presence of N-protected amino acid as metal ligand **71**. They also obtained enantioenriched isoquinolines using amino acid ligand via desymmetrizing C-H activation of prochiral diarylmethylamines with an enantiomeric ratio of up to 98:2 (Figure 24).

An advanced procedure for the synthesis of cyclopropenes was developed via palladium-catalyzed allenylic [4+1] cycloaddition using a planar–chiral ligand by Shao and co-workers [53]. In addition, [4+3] cycloaddition/cross-coupling reaction was observed by replacement of ligand of the palladium catalyst that resulted into the formation of carbocycles bearing 4-spiropyrazolones. Their methodology was proved to be very useful as it provided a facile approach for the formation of [3] dendralenes and led to the discovery of novel compounds with antitumor activity. Cycloaddition of allene acetate **75** with pyrazolone **76** gave spirocyclic product **78** in 82% yield with 93% *ee* using 2.5 mol% [Pd(allyl)Cl]_2_ catalyst and 5.5 mol% of planar–chiral phanePhos ligand **77**. [4+3] Cycloadduct **79** was obtained in 98% yield by loading 5 mol% Pd(cod)Cl_2_ catalyst and 12 mol% triphenyl phosphine ligand (Figure 25).

#### 2.1.6. Platinum and Gold Catalyzed Reactions

A report on the stereo- and regioselective synthesis of indole-based heterocyclic compounds via [3+2] and [2+2] reactions of indolyl allenes was published by Shi and co-workers in 2015 [54]. Different substituted indolyl-allene **1′s** were successfully transformed to a variety of indole-fused heterocycles via Pt and Au-catalysis. Diazabenzo[*a*]cyclopenta[*cd*]azulenes **81** and **82** were synthesized by [3+2] cycloaddition of indollyl allene **80** in the presence of equimolar amount (5 mol%) of PtCl_2_ and [JohnPhosAu]NTf_2_ catalysts in 85% and 96% yields, respectively. Similarly, an eight-membered diazoheterocyclic ring system **83** was also formed in the presence of 5 mol% [IPrAuCl]/AgNTf_2_ by [2+2] *exo*-type cycloaddition in 94% yield. The general mechanism of the [3+2] cycloaddition reaction starts from the generation of metallo-carbon intermediate (A) which after *cis*-addition (in the presence of PtCl_2_) affords compound (B). The formation of Pt-carbene intermediate (C) followed by 1,2-hydride migration affords a targeted product along with the regeneration of catalyst. On the other side, in gold catalyzed reaction intermediate (E) after passing through an intramolecular nucleophilic reaction, tandem cyclization, hydride migration and elimination reaction afforded targeted product (Figure 26).

Construction of methylidene cyclobutane-indoles via Au-catalyzed dearomative [2+2] cycloaddition of *N*-protected indoles with alleneamides and aryloxyallenes was reported by Ocello et al. [55]. Several *N*-protected 2,3-disubstitutive indoles underwent cycloaddition reaction with allenamides and aryloxyallenes to afford different cycloadducts in the presence of (*R*)-DTBM-segphos(AuCl)_2_/AgOTf and [JohnPhosAu(NCMe)]SbF_6_ catalysts. For example, compounds **86** and **88** were synthesized in highest yields (96%) via dearomative [2+2] cycloaddition of oxazolidine substituted allene **85** and *p*-bromophenyloxy allene **87** with N-substituted indole 84 using 5 mol% [Au] catalyst (*R*)-DTBM-segphos(AuCl)_2_/AgOTf and [JohnPhosAu(NCMe)]SbF_6_, respectively (Figure 27).

Triazines act as efficient substitutes for aryl amines and take part in hydroaminomethylation by inserting an amino methyl group to synthesize target molecules. Sun and co-workers reported the Au-catalyzed stepwise [2+2+2] cycloaddition of functionalized allenes with several substituted 1,3,5 triazines to functionalize six-membered N-heterocyclic compounds in high yields (60–96%) [56]. *N*-Heterocyclic compounds **91** (96%) and **93** (89%) were prepared by the cycloaddition of triazine **89** with allenamide **90** and allenoate **92**, respectively using 5 mol% of Ph_3_PAuCl catalyst and NaBAr_F_ (Ar_F_: tetrakis [3,5-*bis*(triflouromethyl)phenyl]borate) (5 mol%) as an additive (Figure 28).

Polycyclic aromatic compounds were synthesized from the cyclization of propargyl carbonates or esters with furan-ynes via gold catalysis by Liu and co-workers [57]. The reaction was initiated with the synthesis of allene by 3,3 rearrangement of propargyl carbonates or esters which underwent a Diels–Alders reaction of furan (IMDAF) to synthesize anthracene derivatives after ring opening of cycloadduct. Using 1,4-furan-yne as substrate, 9-oxygenated anthracene derivatives were formed by aromatization of the cycloadduct while in the case of 1,5-furan-yne, oxa-bridge cleaved in the cycloadduct in association with aryl group 1,2-migration to afford anthracen1(2*H*)-ones. The highest yield (96%) of the functionalized anthracene **95** was obtained from **94** using 5 mol% gold catalyst at 50 °C (Figure 29).

A convenient approach for the synthesis of tetrahydropyrans via the [2+2+2] cycloaddition reaction was reported by research group of López [58]. A highlighted example is presented in Figure 30, showing that the reaction of allenamide **85** with alkene **96** and aldehyde **97** was smoothly processed in the presence of gold catalyst **98** using DCM as unique solvent. As a result, the desired product **99** was obtained in 98% yield (Figure 5). The reaction is highly stereoselective as well as atom economical and covered a wide substrate scope including a variety of aldehydes (aliphatic, aromatic), alkenes (styrene also) and enol ethers or enamides. A similar approach was carried out by this research group in 2017 using gold catalyst **100** to obtain excellent chemo-, regio- and stereoselective tetrahydropyrans and significant results were obtained in this regard (Figure 1) [59].

Marcote et al. reported the use of oxime derivatives as a reaction partner instead of imines in cycloaddition reactions [60]. They reported a straight forward strategy to prepare highly functionalized piperidines and piperidine-containing azabridged carbocycles via gold(Ι)-catalyzed [2+2+2] cycloaddition between allenes and C- and O-tethered oximes. Piperidine derivative **104** was obtained with complete stereoselectivity (*cis* isomer) in an excellent yield (91%) from configurationally *E* pure O-tethered oxime **103** and oxazolidone substituted allene **102**, while cycloaddition of the allenyl ether **102** with *C*-tethered alkenyl oxime **105** resulted in the highest yielded tropane derivative **106** (94%) by loading 5 mol% phosphite gold catalyst **98** in the presence of 4Å MS in DCM (Figure 31). 

### 2.2. Phosphine Catalyzed Cycloaddition Reactions of Allenes

Pyrroloisoquinolines exist in many natural products that exhibit many activities, e.g., (−)-trolline (extracted from the flowers of *T*. *chinensis* Bunge) act as an anti-bacterial agent against respiratory bacteria and an antiviral agent against the influenza virus A and B [61]. Jia et al. reported for the first time the role of isoquinolinium methylides as azomethine ylides in [3+2] cycloaddition with allenes to afford a variety of *N*-heterocycles [62]. The PBu_3_-catalyzed regioselective construction of highly functionalized pyrroloisoquinolines was achieved by dearomatizing the [3+2] addition of several allenones and allenoates with isoquinolinium methylides. Highly substituted pyrroloisoquinoline **109** (87%) was prepared via the dearomative [3+2] annulation of **107** with allenoate **108** using tributylphosphine. A mechanistic approach of this protocol highlights that the addition of phosphine to allene (**108**) first generates intermediate (**A**), which provides intermediate (**B**) after isoquinolinium methylide **107** attack, then intramolecular conjugate addition, sequential *ꞵ*-elimination and isomerization, affording a thermally stable product **109** (Figure 32). 

The first enantio- and diastereoselective construction of 3,2′-pyrrolidinyl-spirooxindole derivatives via the phosphine catalyzed [3+2] cycloaddition of ketimines (isatin derived) with allene esters was reported by Kumar and co-workers [63]. Several phosphine catalysts were screened for the synthesis of [3+2] annulation adduct, spiro-monophosphine, i.e., SITCP was found to be more efficient to afford the desired products stereoselectively. 3,2′-Pyrrolidinyl-spirooxindole derivative (−)-**113** was formed in good yield (88%) and high enatioselectivity (98.7%) using (*R*)−SITCP **112**, which generated the zwitterionic dipole of *α*-cyano-methyl substituted allene ester **111** that underwent [3+2] reaction with *N*-Boc-ketimine **110** (Figure 33).

The monophosphine catalyzed [3+2] cycloaddition of several benzofuranones with allenoates to afford spiro-benzofuranone derivatives was described by Wang et al. [64]. 1-Naphthyl substituted benzofuranone **114** efficiently underwent *γ*-addition [3+2] cycloaddition with allenic ester **115** (R^2^ = H) to synthesize spiro-cycloadduct **116** in 99% yield in the presence of (*R*)-SITCP **112** as chiral phosphine catalyst. Similarly, spiro-benzofuranone **117** was synthesized in 96% yield via asymmetric *α*-addition [3+2] cycloaddition of **114** with *γ*-substituted allenoate **115** (R^2^ = Ph). They also afforded spirooxindoles and spiro-azalactone using the same catalytic system (Figure 34). A zwitterionic intermediate (formed between the reaction of allenoate and phosphine) act as 1,3 dipole that underwent [3+2] cycloaddition with benzofuranone **114** to give phosphorus ylide via *γ*-addition (R^2^ = H) and via *α*-addition (R^2^ = alkyl or aryl group).

An efficient and straightforward synthesis of P-stereogenic phosphines derived from carvone was published by Kwon and co-workers [65]. The synthesized organocatalysts were utilized in the asymmetric synthesis of several pyrrolines via the [3+2] annulation of allenes and imines. When allenoate **108** reacted with *N*-tosylbenzaldimine **118** in the presence of *p*-anisyl phosphines **119**-*S* and **119**-*R*, it resulted in efsevin (a biologically active compound) enantiomers **120**-*S* (92%, 21% *ee*) and **120**-*R* (93%, 84% *ee*), respectively (Figure 35).

Due to the presence of five-membered *N*-heterocycles, a broad range of biologically active compounds, many procedures for the construction of these chiral heterocycles using phosphine catalysts have been described. In this respect, Kramer and Fu presented the synthesis of 2,5-dihydropyrroles via [4+1] annulation of a variety of allenes with different amines catalyzed by spirophosphine catalyst [66]. Among different dihydropyrroles, **124** was synthesized in highest yield (95%) with 89% *ee* by [4+1] annulation of *γ*-substituted allenes **121** with *p*-nitrophenyl sulfonamide **122** in the presence of chiral spirophosphine catalyst **123** at 40 °C (Figure 36).

Gicquel et al. reported the preparation of phosphahelicene bearing an isopinocampheyl group on phosphorus and utilized them as organocatalyst in [3+2] cyclization of aryl/alkylidenemalononitriles with *γ*-substituted allenes [67]. Several cyclopentene derivatives were synthesized in excellent yields and high diastereoselectivities with up to 97% in enantiomeric excess. Particularly, when arylidenemalononitrile **125** underwent [3+2] cyclization with benzyl 6-phenylhexa-2,3-dienoate **126** in the presence of phosphahelicene **127** (10 mol%), the highest yield of cyclopentene **128** (92%) was obtained with good enantioselectivity (96%) and high diastereoselectivity (>95:5 *dr*) (Figure 37). 

The addition/cycloaddition domino reactions of *β*′-acetoxy allenoates with 2-acyl-3-methyl-acrylonitriles and 2-acyl-3-(2-pyrrole)-acrylonitriles to afford 2-oxabicyclononanes and cyclopentapyrrolizines, respectively, was reported by Tong and co-worker [68]. 2-Oxabicyclo [3.3.1]nonanes were synthesized through *β*′-addition/[4+4] cycloaddition of allenoates, in which *β*′C and *γ*C served as a 1,4-dipole and *β*′C acted as electrophilic center, with 2-acyl-3-methyl-acrylonitriles, while *γ*-addition/[3+2] cycloaddition was observed in the synthesis of cyclopenta[a]pyrrolizines in which *β*C and *β*′C of allenoate served as 1,3-dipole and *γ*C displayed dual electrophilicity. For example, **131** and **133** were synthesized from allenoate **129** via phosphine-catalyzed addition/cycloaddition reactions with 2-acyl-3-methyl-acrylonitrile **130** and 2-acyl-3-(2-pyrrole)-acrylonitrile **132** in 88% and 95% yields, respectively (Figure 38).

### 2.3. Miscellaneous 

An effective report on the construction of 4*H*-pyran derivatives via [4+2] cycloaddition of 2,3-dioxopyrrolidines with allene ketones using *cinchona* alkaloid-derived amine as the catalyst was published by Xu and co-workers [69]. Several catalysts were used for the formation of 4H-pyrans but *cinchona* alkaloid-derived amine **136** gave excellent yields (59–90%) and high enantioselectivities (up to 97% *ee*). 2,3-Dioxopyrrolidine with *m*-bromophenyl **134** and allene ketone bearing phenyl substituent **135** gave the highest yield (90%) of the 4H-pyran-fused pyrrolin-2-one **137** with 92% *ee* (Figure 39).

Conner et al. reported a chiral Lewis acid (**140**) catalyzed [2+2] cycloaddition reaction between allenoate and alkene to achieve excellent enantioselectivity of the corresponding products [70]. The methodology covered a wide substrate scope that was equally suitable for inactivated alkenes. However, trisubstituted alkenes and *α*- or *γ*-substituted allenes gave the desired products with low selectivity via this protocol. A highlighted example of this protocol is depicted in Figure 40. When alkene **138** was treated with allene **139** in the presence of 20 mol% catalyst **140**, as a result, a targeted product **141** was obtained in 82% yield with 98:2 er and 7:1 *E*:*Z*.

Another report on the synthesis of cyclobutane derivatives via the intramolecular [2+2] cycloaddition of alkenes and allenoates was published by Xu et al. [71]. Among different substrates, allene **142** gave cycloadducts **144**-*E*, **144**-*Z* in highest yield (70%) and good enantioselectivity (1:20 *E*:*Z*) by loading 20 mol% chiral oxazaborolidine catalyst **143** (Figure 41). 

Miao and co-workers reported the construction of tetrahydropyrano [2,3-c]pyrazole derivatives through the regioselective [4+2] cycloaddition of *α,β*-unsaturated benzylidenepyrazolones with allene ketones or *α*-methyl allene ketones using nitrogen-bearing Lewis base [72,73,74]. They utilized quinine **147** and DMAP as Lewis base catalysts for the construction of tetrahydropyrano [2,3-c]pyrazoles **148**–**150** which resulted in two different adducts, *α* and *γ*, respectively. Both *α*- and *γ*-adducts were synthesized in 99% yields from benzylidenepyrazolone **146** and substituted allene ketone **145** using quinine **147** (20 mol%) and DMAP, respectively (Figure 42). First, a zwitterionic intermediate formed as a result of the reaction between allene ketone and Lewis base catalysts (quinine and DMAP) that after several steps led to the formation of *α*-adduct (in case of quinine) and *γ*-adduct (in the presence of DMAP). 

Liu et al. reported an effective and green method for the preparation of cyclobuta[*a*]naphthalen-4-ols that took place through different approaches including: (1) [2+2] cycloaddition, (2) SO_2_ insertion, (3) 1,4-addition, (4) diazotization and (5) tautomerization [75]. They reported straightforward synthesis of novel cyclobutanaphthalen-4-ols by first presenting a multicomponent bicyclization strategy. Allene-ynes/benzene-linked allene-yne esters underwent a [2+2] cycloaddition reaction with aryldiazonium tetrafluoroborates, which after insertion of SO_2_, resulted in desired products. Aryldiazonium tetrafluoroborates **152** bearing *p*-ethoxy group was reacted with benzene-linked allene-yne ester **151** to obtain 94% yield of the product **154** via intermediate **153** in the presence of DABSO, 1,2-dichloroethane (DCE) and *p*-ethoxy benzene (Figure 43).

Kapur et al. developed a thermal reaction of 3-(N-aryliminomethyl)chromones with substituted 2,3-butadienoates on refluxing in dry benzene that resulted into the synthesis of some novel compounds by reorganization of [2+2] cycloadducts [76]. For example, when 3-(N-aryliminomethyl)chromone **155** reacted with ethyl 2,3-butadienoate **156** (when R^3^ = H) or ethyl 4-phenyl-2,3-butadienoate **156** (when R^3^ = Ph), only cycloadduct **157** was formed in 70–79% yield. However, when **155** was reacted with ethyl 2,3-pentadienoate **156** (when R^3^ = Me) in similar conditions, compounds **158** and **159** were formed in 47–52% and 32-40% yields, respectively (Figure 44). 

Chen et al. presented the synthesis of chiral benzylic sulfones and 4-substituted chromans via the dynamic kinetic resolution (DKR) of 2-sulfonylalkyl phenols with allenic esters and formal [4+2] cycloaddition of 2-(tosylmethyl)sesamols or 2-(tosylmethyl)-naphthols with allenic esters, respectively [77]. *o*-Quinone methide intermediate was generated in both, (1) the racemization of 2-sulfonylalkyl phenols followed by asymmetric addition catalyzed by cinchonine-derived catalyst and (2) the enantioselective [4+2] cycloaddition reaction. The highest yielded benzylic sulfone **163** (79%, 87% *ee*) and 4-substituted chroman **166** (90%, 97% *ee*) was obtained from the reaction of allenic ester **160** with 2-sulfonylalkyl phenol **161** and 2-(tosylmethyl)-naphthol **164** by using cinchonine-derived catalyst **162** and *cinchona* alkaloid catalyst **165**, respectively (Figure 45).

Garg and co-workers studied azacyclic allenes and heteroatom bearing cyclic allenes, which could not gain enough attention by synthetic chemists [78]. They reported (1) the synthesis of azacyclic allene precursors in mild reaction conditions, (2) the trapping of the desired cyclic allenes in the Diels–Alder reaction to afford functionalized piperidine products and (3) [3+2] cycloaddition of heterocyclic allenes. They also proved that stereochemistry of the enantioenriched substrates transferred via stereochemically defined azacyclic allene intermediate to Diels–Alder products. Silyl triflate **167** was prepared (starting from 4-methoxypyridine) as a precursor of azacyclic allene **169** that trapped in the [3+2] cycloaddition reaction with 3,4-dihydroisoquinoline 2-oxide **168** in the presence of CsF to afford tetracyclic product **170** in a quantitative yield with 5.3:1 *dr* (Figure 46). 

A one pot three component reaction of allenic ketones/allenoates, amines and enones was reported by Feng et al. to synthesize cyclohexa-1,3-dienes (in the absence of oxidant) and 2-aminobenzophenones/benzoate derivatives (in the presence of oxidant) at elevated temperature (120 °C) in dioxane [79]. The synthesis of the desired products proceeded with the synthesis of the enaminone intermediate by the nucleophilic addition of allenic ketone with amine preceded by Michael addition which underwent catalyst/base-free [3+3] annulation with enone. Electron donating substituents on the phenyl ring of allenic ketones resulted in better yields as compared to phenyl bearing electron withdrawing groups. For example, highly functionalized cyclohexa-1,3-diene **174** and 2-aminobenzophenone **175** were obtained from allenic ketone **171**, amine **172** and enone **173** in highest yield (86% and 79%, respectively) (Figure 47).

Ueda and co-workers presented the synthesis of cyclopentene/cyclobutane-annulated fullerenes via base-catalyzed [3+2] and [2+2] cycloaddition of 1,3-bifunctional allenes (generated in situ) in *ortho*-dichlorobenzene (ODCB) [80]. The synthesis of cyclopentene-annulated fullerenes was obtained from Et_3_N-catalyzed [3+2] cycloaddition of propiolic acid esters and 1,2-diaryl-1,2-diketones with C_60_. Among several substituted 1,2-diaryl-1,2-diketones, 4,4′-difluorobenzil **177** (Ar = 4-F-C_6_H_4_) was proved to be very reactive with propiolic acid ester **178** and C_60_ **176** that resulted in the highest yielded cyclopentene-annulated fullerene **179** (46%). Similarly, cyclobutane-annulated fullerenes were synthesized on a flow packed-bed reactor combined with silica bearing tertiary amine. 1,3-Bifunctional allene was synthesized in packed-bed reactor by silica-supported tertiary amine **180** that afforded the desired [2+2] cycloadducts after reacting with C_60_ in a tubular reactor. In the case of cyclobutane-annulated fullerenes, 3,3′-dimethoxybenzil **177** (Ar = 3-OMe-C_6_H_4_) was proved highly reactive with propiolic acid ester **178** and C_60_
**176**, that gave a 41% yield of the product **181** (Figure 48). Moreover, it has been recently discovered that the cycloaddition reactions, i.e., 32CA, proceed swiftly by involving C20 fullerenes as there has been great attraction found of dienes towards C20 fullerenes [81].

Shi and co-workers proved that allenes could act as analogous to alkynes in the building of bioactive spiro[indoline-3,2′-pyrrole] with excellent yields and good enantioselectivities [82]. They described the usage of allenes instead of alkynes to afford enantioselective spiro[indoline-3,2′-pyrrole] derivatives via catalytic asymmetric isatin-involved 1,3-dipolar cycloaddition (1,3-DC). They reported asymmetric 1,3-DC of allenes with azomethine ylides (derived from isatin) to afford enantioenriched spiroindolinepyrroles. An unexpected formation of spirooxindole with an intraannular carbon double bond was also observed. *Bis*-phosphoric acid (*Bis*-PA) **185** (15 mol%) efficiently catalyzed 1,3-DC and assembled isatin **182**, 2,3-allenoate **183** and amino-ester **184** afforded desired product **186** in 65% yield with 93% *ee* along with the formation of compound **187** (Figure 49). 

Yu and co-workers developed a metal-free approach towards the construction of pyrrolidines via the cycloisomerization and intramolecular [4+3] cycloaddition of allene-alkynylbenzenes, respectively mediated by Brønsted acids (TfOH, HBF_4_ or Me_3_OBF_4_) [83]. The synthesis of pyrrolidine derivatives was proceeded via the formation of vinyl cation by the reaction of alkyne with allylic cation (generated from allene), grabbed by triflate (TfO) anion to afford the desired product. In excess acid, the cycloisomerization product underwent Friedel–Crafts reaction to attain seven membered rings by TfOH-mediated intramolecular [4+3] cycloaddition reaction. Pyrrolidine derivative **189** was obtained in 85% yield from substrate **188** at room temperature using 1.1 equivalents of TfOH while [4+3] cycloadduct **190** was synthesized at 60 °C in the presence of excess TfOH (10 equivalents) in highest yield (94%) (Figure 50). This protocol can also be used to synthesize F-incorporated products using HBF_4_ or Me_3_OBF_4_ as the fluoro source.

Liu and co-workers described the preparation of 1-sulfonyl-trifluoromethyl allenes and their utilization in [3+2] cycloaddition reaction with nitrones to afford a series of trifluoromethylated isoxazolidine derivatives without using any catalyst [84]. Starting with 2-bromo-3,3,3-trifluoropropene **191**, a variety of substituted allenes **192** were synthesized in 67–88% yields using various aldehydes or ketones. The synthesized 1-sulfonyl-trifluoromethyl allenes **192** underwent [3+2] cycloaddition with different substituted nitrones **193** that resulted in the formation of trifluoromethylated isoxazolidines **194** in excellent yields (86–94%) (Figure 51).

An easy and simple approach towards the synthesis of strained polycyclic compounds without using any catalyst was reported by Cheng et al. that involved an Ugi/Himbert arene/allene Diels–Alder cycloaddition reaction [85]. The desired strained polycycles were synthesized via a multicomponent reaction of several substituted aldehydes/ketones, aniline, isocyanide and allenic acid in methanol. The highest yielded (67%) polycycle **198** was synthesized using benzaldehyde **18**, aniline **195**, isocyanide **196** and allenic acid **197** (Figure 52). Their synthetic approach proceeded through the formation of a Ugi adduct that underwent a Diels–Alder reaction between the terminal allene and aromatic ring. This terminology has some advantages including (1) wide substrate scope, (2) no need for protection and (3) no transformation of acid into acyl chloride. 

Arai and Ohkuma reported the [2+2] photochemical cycloaddition of substituted indole derivatives to afford stereoselective methylenecyclobutane-fused indolines in the presence of aromatic ketones as sensitizers irradiated by a high pressure Hg-lamp by Pyrex [86]. This protocol is very significant as it affords heterocyclic compounds via photochemical reaction without using any catalyst. Among different ketones, 3,4-dimethoxyacetophenone was more effective to synthesize all-*cis*-fused methylenecyclobutane-type compounds in good yields. For example, methylenecyclobutane-type product **201** was synthesized in 72% yield accompanied by 14% terminal alkyne **202** in the presence of 50 mol% 3,4-dimethoxyacetophenone **200** under irradiation. However, only [2+2] cycloadduct **203** was formed from trisubstituted allene **199**, suggesting an internal transposition of the terminal hydrogen of allene to C3 of indole resulted in alkyne moiety (Figure 53). 

An efficient diastereoselective formation of chiral tetrahydrofuran was reported by Wang et al. [87]. They found *α*-allenic amides as suitable dipolarophile in the [3+2] cycloaddition with vinyl epoxides using Pd-catalyst and *N*-heterocyclic carbene (NHC) as ligands which resulted in tetrahydrofuran derivatives having three functionalities; (1) tetrasubstituted enolether, (2) monosubstituted alkene and (3) amide. For example, tetrahydrofuran derivative **207** was synthesized in an excellent yield (99%) with good enantioselectivity (94% *ee*) from the [3+2] addition of allenic-amide **204** with vinyl epoxide **205** using [Pd(η^3^-C_3_H_5_)Cl]_2_ (5 mol%) catalyst and NHC precursor **206** (11 mol%) as ligand (Figure 54).

### 2.4. Synthesis of Natural Products

Allenes act as unique building blocks in synthetic organic chemistry for the construction of complex bioactive compounds and natural products in a straightforward manner. Many reports on the construction of natural products via the cycloaddition of allenes have been published using different transition metal complexes. 

#### 2.4.1. Synthesis of Guaiane Family

Evans and co-workers described the stereoselective synthesis of tri- and tetrasubstituted exocyclic alkenes via carbocyclization of several alkynylidenecyclopropanes (ACPs) with activated and inactivated allenes [88]. Their synthetic protocol for the formation of substituted exocyclic olefins was well suited for the synthesis of the guaiane family of sesquiterpenes via distal insertion of disubstituted allenes into ACPs. The desired carbon skeleton of guaiane **210** was constructed by carbocyclization of malonate tether ACP **208** with activated allene **209** using [Rh(cod)Cl]_2_ (5 mol%) and triphenylphosphite (P(OPh)_3_) (30 mol%) in *p*-xylene at 120 °C (Figure 55).

#### 2.4.2. Synthesis of (−)-Vindoline and (+)-4-*epi*-Vindoline 

(−)-Vindoline **211** is a biologically active clinic alkaloid, derived from the leaves of *Cantharanthus roseus*, that acts as starting material for the synthesis of natural products such as vincristine and vinblastine **213**. Its 4-*epimer*, (+)-4-*epi*-vindoline **212, is** used to synthesize (+)-4-*epi*-vinblastine **213**. The structure of (−)-vindoline **211** and (+)-4-*epi*-vindoline **212** consists of two five-membered and three six-membered fused rings (Figure 2) [89].

Boger and co-workers reported an efficient synthetic protocol in which the intramolecular [4+2]/[3+2] cycloaddition of 1,3,4-oxadiazoles was initiated by allene dienophile that led to the formation of a pentacyclic core system of vindoline **211** and its C4 epimer **212** [90]. Initial cycloadduct **215** was formed as a result of a Diels–Alder reaction between 1,3,4-oxadiazole and allene **214** that underwent nitrogen loss to afford carbonyl ylide **216**. Cross-conjugated 1,3-dipole **216** underwent indole endo [3+2] cycloaddition reaction which resulted in single diastereomer **217** in 92% yield, and after many steps, formed ketone **218**. (−)-Vindoline **211** and (+)-4-*epi*-vindoline **212** were formed after several steps from ketone **218**, and later transformed to 4-*epi*-vinblastine **213** in a single step with 44% yield (13 step total synthesis) (Figure 56). 

#### 2.4.3. Formal Synthesis of (−)-Galanthamine

(−)-Galanthamine **219** is an alkaloid having 3,4-cyclohexenol skeleton, belongs to the Amaryllidaceae family, and was accidentally discovered in the early 1950s and initially used to treat poliomyelitis. It has been recently approved for the treatment of Alzheimer’s disease as it acts as a reversible competitive inhibitor of acetyl cholinesterase (Figure 3) [91].

Liu and Yu developed a useful methodology for the synthesis of 2-methylidene-3,4-cyclohexenones via Rh-catalyzed [5+1] cycloaddition of ACPs with carbon monoxide [92]. Their synthetic protocol was utilized for the formal synthesis of (−)-galanthamine **219** from cycloadduct **221** prepared from the [5+2] cycloaddition of ACP **220** with CO using [Rh(CO)_2_Cl]_2_ (5 mol%). Alcohol **223** was formed in 79% yield with 97% *ee* using CBS reduction, which after several steps formed aldehyde **224**. A reduction of aldehyde **224** with sodium borohydride gave Brown’s intermediate **225** that eventually transformed to (−)-galanthamine **219** using a previously reported method [93] (Figure 57).

#### 2.4.4. Diastereoselective Synthesis of Diquinanes and Triquinanes

Polyquinanes (class of carbocyclic frameworks) are part of many natural products such as steroids and terpenoids that contain condensed five-membered rings. Waihoensene **226** (a tetracyclic diterpene) was first isolated in 1997 by Weavers and co-workers from New Zealand podocarp *Podocarpus totara var waihoensis* (Figure 4) [94].

Yang and co-workers in 2017 developed a diastereoselective synthesis of a [3.3.0] bicyclic system via an intramolecular [3+2] cycloaddition of *α*,*β*-unsaturated aldehydes or esters and allenes initiated by thiyl radical [95]. Several substituted diquinanes were synthesized through the intramolecular [3+2] cycloaddition reaction of allene in the presence of PhSH and 2,2′-azobis (2,4-dimethylvaleronitrile) (ABVN) as thiyl-radical initiator at 70 °C in 39–73% yields. Angular fused triquinane **228** was also synthesized from **227** (prepared from Stoltz’s Pd-catalyzed decarboxylative allylation with 92% *ee*) in 30% yield over two steps with 92% *ee* that could lead to the formation of waihoensene **226** (Figure 58).

#### 2.4.5. Synthesis of *ent*-[3]-Ladderanol

*ent*-[3]-Ladderanol **229** belongs to ladderane family that was first isolated from annamox bacteria in 2002 and consists of fused cyclobutene rings. Ladderanes are very useful in biological systems as they increase the barrier for the diffusion of toxic substances by incorporating into the lipid bilayer of cell membranes (Figure 5) [96,97].

Brown and co-workers described the enantioselective preparation of *ent*-[3]-ladderanol starting from easily available alkyne **230** and epoxide **231** in 14 steps [98]. Their synthetic strategy provided [4.2.0]-bicycles via the chirality transfer [2+2] cycloaddition of alkenes with allenic ketones. First, *β,γ*-alkynyl ketone **232** was synthesized by the addition of alkyne **230** and epoxide **231** followed by oxidation with Dess–Martin periodinane, that was then enantioselectively isomerized to allene **234** in the presence of thiourea catalyst **233**. The required [4.2.0]-bicycle **235** was synthesized via a chirality transfer [2+2] cycloaddition reaction by adding MeNO_2_ and Bi(OTf)_3_ (Figure 59).

[4.2.0]-Bicycle **235** was utilized to synthesize compound **236** after several steps that underwent a [2+2] cycloaddition reaction with cyclopentenone to afford **237** that gave *ent*-[3]-ladderanol **229** in 51% yield over three steps (Figure 60).

#### 2.4.6. Synthesis of Chiral Carbocyclic Nucleosides

Carbocyclic nucleosides, in which a methylene group is replaced by one oxygen, are biologically very important, for example, aristeromycin **238** (a natural carbocyclic nucleoside) acts as antiviral agent, (1*R*,4*S*)-carbovir **239** (chiral carbocyclic nucleoside) is a potential HIV-1 inhibitor, while entecavir **240** and abacavir **241** have been approved by food and drug administration (FDA) to treat viral infections (Figure 6) [99,100,101].

Gao et al. employed *N*-heteroaromatic-substituted acrylates in [3+2] cycloaddition with 2,3-butadienoates to afford several analogues of carbocyclic nucleosides with a C=C bond and a quaternary carbon catalyzed by chiral phosphine [102]. Different chiral phosphine catalysts were screened for a [3+2] annulation reaction; among them, spirocyclic phosphine catalyst with a bulky P-aryl substituent gave a good yield and high enantioselectivity using 2-naphthol. The reaction protocol was found useful as *α*-benzimidazole substituted acrylates and *α*-purine-containing disubstituted acrylates could participate too in phosphine-catalyzed [3+2] cycloaddition reaction. Carbocyclic nucleoside analogue **245** was obtained in highest yield (90%) with 93% enantioselectivity from *α*-purine-substituted acrylate **242** and 2,3-butadienoate **243** by using catalyst **244** (20 mol%) and 20 mol% 2-naphthol in DCM at 0 °C (Figure 61).

#### 2.4.7. Synthesis of Hebelophyllene E 

Hebelophyllene E **246** is one of eight members of *cis*-fused caryophyllene-type sesquiterpenes that were isolated from *Hebeloma longicaudum* (an actomycorrhizal fungus) in the late 1990s, and structurally consist of *geminal* dimethyl cyclobutane (Figure 7) [103]. 

An enantioselective synthesis of chiral *gem* dimethylcyclobutane derivatives was reported using a novel oxazaborolidine catalyst in [2+2] cycloaddition of allenoates and alkenes by Wiest et al. [104]. They developed the first synthesis of hebelophyllene E **246** (sesquiterpene) and assigned the relative configuration to the side chain by synthesizing *epi*-*ent*-hebelophyllene E. For this purpose, they synthesized fully functionalized alkene **249** from enantiopure acetate **248**, from a previously reported method by Wessjohann [105] starting from compound **247** using amino lipase PS, by (1) the addition of vinylmagnesium bromide and (2) acetonide protection in 59% yield and >99:1 diastereoselectivity (Figure 62).

Alkene **249** underwent [2+2] cycloaddition with benzyl allenoate **243** using oxazaborolidine catalyst *ent−***250** to afford (*Z*)−**251** in 52% yield with >99:1 *er* and >99:1 *dr*. Cyclobutane *cis−***253** was obtained using *ent−***252** ligand in 99% yield and 89:11 *dr* which, after several steps, formed hebelophyllene E **246** in 99:1 *er* and 99:1 *dr* (Figure 63).

## 3. Conclusions

Allenes have unique cumulative system with two contiguous carbon–carbon double bonds which make them a versatile synthetic unit in organic chemistry. This review highlights the use of substituted allenes in several metal catalyzed cycloaddition reactions for the straightforward synthesis of carbo-/heterocycles in one-step considering their chemo- regio- and stereoselectivity in view. A number of transition metals including platinum, gold, rhodium, palladium, nickel, cobalt, titanium and phosphine have been used to carry out these conversions effectively. Furthermore, the synthetic applications of these protocols towards the synthesis of natural products have also been described briefly. Hopefully, this review and the cited examples will provide a great opportunity for the synthetic chemists to develop novel chiral catalytic systems for the cycloaddition reactions of allenes. Though a significant effort has been made in this area, significant improvement is still required, especially for the stereoselective synthesis of natural products and other pharmaceutically important drugs via these types of cycloaddition reactions, which is expected in the near future.

## Data Availability

Not applicable.

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
