# Peer review of "Unveiling the Chemistry and Synthetic Potential of Catalytic Cycloaddition Reaction of Allenes: A Review"

_molecules, 2023, doi:10.3390/molecules28020704_

Round 1
Reviewer 1 Report
-------------------------------
General remarks:
(1) Only the part of analysed reactions can be considered as cycloaddition. The fundamental rules of cycloaddition reaction are excellent known from many decades. In particular, in the course oc cycloaddition processes (a) any sigma bonde are not break, (b) new sigma bonds are formed from the electrons derived from the break pi-bonds. (etc.)
So, the title of the paper and discussion of some examples should be substantially changed.
(2) Informations regarding to the mechanistic aspects of cycloaddition processes should be fundamentally corrected in the introduction.
(2a) Firstly, terms "concerted" and "pericyclic" are completly outdated and should be removed [Molecules 21, 1319 (2016)]. According to last discoveries and the actual state of knowledge, it is known, that any cycloaddition reactions are not realised vie these-type hypothetical mechanism.
(2b) Next, in generall, (4n+2)-pi-electron cycloaddition processes in most examples are realised via one-step mechanism. This mechanism can characterised by different ways of electron density reorganisation. In any cases hovewer there are not "concerted" ("pericyclic") process.
(2c) Next, in generall, (4n)-pi-electron cycloaddition processes in most examples are realised via stepwise mechanism with the participation of zwitterionic or biradical intermediates.
(2d) Next, some (4n+2)-pi-electron cycloaddition processe can realised via stepwise mechanisms. The possibility of these-tpe mechanisms are critical discussed in the last time [Symmetry, 13, 1911 (2021), Organics, 1, 49 (2020)].
-------------------------------
Other, minor remarks:
Figure 1 is not adequate and should be removed.
Caption of the Scheme 1
In the course of [2+2] cycloaddition, the four-membered cyclic system is formed.
Scheme 3 and discussion
This is not formation of 6-membered heterocyclic sysytem, but functionalisation reaction of starting material which include the 6-membered heterocyclic system.
Scheme 5
The reaction sequence should be presented in details.
Caption of the Scheme 44
In the course of [2+2] cycloaddition, the four-membered cyclic system is formed.
Caption of the Scheme 46.
The name of the nitrone should be precised. Next, this is a example of [3+2] cycloaddition reaction.
Page 34.
Kinetic and mechanistic aspects of the different type cycloaddition processes with the particiupation of fullerense molecular segments was recently discussed in detail [Scientiae Radices, 1, 46 (2022)].
Scheme 51
C-aryl-N-alkyl nitrones exhibit allways Z-configuration - not E.
Author Response
Respected reviewer, as per suggestion, appropriate changes have been made in the manuscript.
Thank you very much.

Reviewer 2 Report
The authors discuss the most relevant publications on transition metal and phosphine catalyzed cycloaddition reactions of allenes and highlight their synthetic applications toward the synthesis of natural products. The topics is interesting and attractive, however, there are some references such as (1) and its cited references which are relevant to the topics should be cited.
Some typos should be corrected, such as in Line 46,“tom” and in Line 514, “1379”
Reference: (1) M. M. Afonso and J. A. Palenzuela, “1,3-Dipolar Cycloadditions Involving Allenes: Synthesis of Five-membered Rings” Current Organic Chemistry, 2019, 23, 3004-3026, and the references cited.
Author Response

(The authors gave the same response as above.)

Reviewer 3 Report
Authors describe the advances made in catalytic cycloaddition reaction of allenes since 2015 in a review, showing the more recent milestones reached in that field. Although I find the idea of a recent reviewing of the literature about cycloaddition reactions very attractive, authors must take over several issues found in the manuscript.
First, the title of the manuscript mention metal catalyzed whereas the whole manuscript describes the employment of metals, phosphine, and organic compounds as catalyst. Thus, I would suggest to authors to change the title to something like: “Unveiling the Chemistry and Synthetic Potential of Catalytic Cycloaddition Reaction of Allenes.”
Additionally, authors must review the formulas all over the manuscript and make sure that all the subscripts are spelled correctly rather than just put them in lower case. For instance, in line 72 they wrote PtCl2 instead of PtCl2. This mistake is repeated all over the manuscript. Moreover, I recommend to the authors to check the red book from IUPAC to spell right the formulas all over the text and schemes. For example, in line 181, the Rh complex is formulated with charges and that is wrong.
Some of the mistakes I found are listed below:
Line 46: atom
Line 119: López
Line 121: as the only solvent= as unique solvent
Line 153: no brakes, “1,7-H-shift”
Scheme 7: check the spelling of Cp*
Line 170: as the result
Line 268: as pre-catalyst
Furthermore, the results presented in the review need to have some order, in the first epigraph is “metal-catalyzed” and it is split in metals, however, there is not an order in that, usually those description are made following the group order in the periodic table. I would recommend to the author to regroup the milestones based on the group of the periodic table, form left to right or vice versa.
Finally, an English language and style revision is highly recommended.
Once the authors overcome to the suggestions made, the paper might be a good candidate to be published in molecules
Author Response

(The authors gave the same response as above.)

Round 2
Reviewer 1 Report
Authors considered my remarks and improved the paper accordingly. The paper require at this moment only one small correction:
Please to correct the configuration of C-aryl-N-alkyl nitrones. In cited reference (Tetrahedron Lett. 2017, 58, 3377) this configuration is described wrongly. C-aryl-N-alkyl nitrones exhibit allways Z-configuration - not E. This amendment should be introduced, because the review work should be critical analysis, not a simple statement of content from source work.
Author Response
Point 1: The paper require at this moment only one small correction:
Please to correct the configuration of C-aryl-N-alkyl nitrones. In cited reference (Tetrahedron Lett. 2017, 58, 3377) this configuration is described wrongly. C-aryl-N-alkyl nitrones exhibit allways Z-configuration - not E. This amendment should be introduced, because the review work should be critical analysis, not a simple statement of content from source work.
Response 1: Worthy reviewer, we are thankful for your suggestions. We have now changed the configuration of nitrone as per your suggestion.
Reviewer 3 Report
I really apreciate the effort made by the authors taking into account all changes suggested. Now, I truly belive the manuscript is perfectly suitable for publishing.
Author Response
Point 1: I really apreciate the effort made by the authors taking into account all changes suggested. Now, I truly belive the manuscript is perfectly suitable for publishing.
Response 1: Thank you very much respected reviewer. We highly appreciate your suggestions that helped us improve our manuscript.